# Human Monkeypox: A Comprehensive Overview of Epidemiology, Pathogenesis, Diagnosis, Treatment, and Prevention Strategies

**DOI:** 10.3390/pathogens12070947

**Published:** 2023-07-18

**Authors:** Diana Emilia Martínez-Fernández, David Fernández-Quezada, Fidel Antonio Guadalupe Casillas-Muñoz, Francisco Josué Carrillo-Ballesteros, Ana Maria Ortega-Prieto, Jose M. Jimenez-Guardeño, Jose Angel Regla-Nava

**Affiliations:** 1Transdisciplinary Institute of Research and Services (ITRANS), University of Guadalajara, Zapopan 45150, Mexico; diana.martinez@academicos.udg.mx (D.E.M.-F.); fidel.casillas@academicos.udg.mx (F.A.G.C.-M.); josue.carrillo@academicos.udg.mx (F.J.C.-B.); 2Department of Neurosciences, University Center for Health Science (CUCS), University of Guadalajara, Guadalajara 44340, Mexico; david.fernandez@academicos.udg.mx; 3Department of Microbiology, University of Málaga, 29010 Málaga, Spain; ana.ortega@uma.es (A.M.O.-P.); jose.jimenez@uma.es (J.M.J.-G.); 4Instituto de Investigación Biomédica de Málaga y Plataforma en Nanomedicina-IBIMA Plataforma BIONAND, 29590 Málaga, Spain; 5Department of Microbiology and Pathology, University Center for Health Science (CUCS), University of Guadalajara, Guadalajara 44340, Mexico

**Keywords:** monkeypox (MPX), smallpox, Orthopoxvirus, zoonotic disease, public health emergency

## Abstract

Monkeypox virus (MPXV) is an emerging zoonotic virus that belongs to the Orthopoxvirus genus and presents clinical symptoms similar to those of smallpox, such as fever and vesicular–pustular skin lesions. However, the differential diagnosis between smallpox and monkeypox is that smallpox does not cause lymphadenopathy but monkeypox generates swelling in the lymph nodes. Since the eradication of smallpox, MPXV has been identified as the most common Orthopoxvirus to cause human disease. Despite MPXV being endemic to certain regions of Africa, the current MPXV outbreak, which began in early 2022, has spread to numerous countries worldwide, raising global concern. As of the end of May 2023, over 87,545 cases and 141 deaths have been reported, with most cases identified in non-endemic countries, primarily due to human-to-human transmission. To better understand this emerging threat, this review presents an overview of key aspects of MPXV infection, including its animal reservoirs, modes of transmission, animal models, epidemiology, clinical and immunological features, diagnosis, treatments, vaccines, and prevention strategies. The material presented here provides a comprehensive understanding of MPXV as a disease, while emphasizing the significance and unique characteristics of the 2022 outbreak. This offers valuable information that can inform future research and aid in the development of effective interventions.

## 1. Introduction

Human monkeypox is a zoonotic infection caused by the monkeypox virus (MPXV), which is a double-stranded DNA virus belonging to the *Orthopox* genus in the *Poxviridae* family [1]. This viral infection was first identified in captive monkeys in 1958 in Copenhagen, Denmark, as the cause of a non-fatal pox-like illness [2]. However, the first confirmed case of human infection reported was described in a child from the Democratic Republic of the Congo (DRC) in 1970 [3]. For several decades, cases of human monkeypox have been reported annually in western and central Africa, with most of the cases being identified in the DRC, indicating that the disease has remained endemic in these regions. Nonetheless, the recent and sudden emergence of monkeypox cases in non-endemic countries implies that there could have been undetected transmission for an unknown period of time [4].

Traditionally, MPXV has been genetically classified into two main clades: the Congo Basin (central African) clade, or Clade I, which is considered to be more virulent and associated with a higher case fatality rate (>10%), and the West African clade, or Clade II, which is associated with a lower case fatality rate (<1%) [5]. However, recent bioinformatic analyses using MXPV genomes from isolated samples suggested the existence of a third clade, referred to as Clade IIb or Clade III [6,7].

The recent spread of monkeypox cases in non-endemic countries is likely due to the absence of Orthopoxvirus immunity among these populations. This can be attributed to the cessation of smallpox vaccination after the disease was declared eradicated. Currently, monkeypox disease is considered the most significant Orthopoxvirus infection in humans [8]. 

In July 2022, the WHO declared that the mpox outbreak represented a Public Health Emergency of International Concern (PHEIC) [9]. This new term—“mpox”—for monkeypox disease was recommended by the World Health Organization (WHO) in November 2022 [10]. However, 10 months later, the WHO declared an end to the mpox public health emergency, on 11 May 2023 [11]. 

The initial symptoms of mpox include back pain, headache, and a rash. In more severe cases, the disease can cause pneumonitis, encephalitis, or secondary bacterial infections [12]. Lymphadenopathy is a hallmark clinical manifestation of mpox, a distinctive feature frequently used to differentiate it from other contagious diseases such as chickenpox, measles, and smallpox [12].

This review provides an overview of the recent advances in our understanding of human monkeypox infection, with particular emphasis on the significant re-emergence in 2022. Based on the available data, this article examines key aspects of the disease, including its public health importance, animal reservoirs, mode of transmission, surveillance, diagnosis, and vaccination.

## 2. Public Health Importance

Monkeypox comprises a large brick-shaped virus (200–250 nanometers) that is surrounded by an envelope containing a linear double-stranded DNA genome [13]. The Orthopoxvirus genus represents a potential threat to humans due to its wide host spectrum. This genus includes the smallpox virus or variola virus (VARV), which was declared eradicated by the WHO in 1980, after killing 300–500 million people in the 20th century [14]. Interestingly, MPXV has been described as being related to the smallpox virus [15]. Other viruses in the Orthopoxvirus genus that can infect humans include the cowpox virus (CPXV) and vaccinia virus (VACV), which are utilized in the smallpox vaccine. Additionally, there are other human pathogenic Orthopoxviruses, such as the Abatino virus [16], Akhmeta virus [17], and Alaskapox virus [18], which have recently been described in the United States of America (USA), Georgia, and Italy.

The main animal vectors of Orthopoxviruses are domestic animals, food animals, and rodent species of different habitats [13]. The molecular evolution and diversity of these viruses, together with their broad host range, increase the possibility of the re-emergence and emergence of outbreaks caused by Orthopoxviruses [19].

The recent re-emergence of monkeypox was first reported on 7 May 2022, in the United Kingdom, followed by Portugal on 17 May. On 20 May, Australia, Germany, the Netherlands, and France confirmed their first cases, while the Czech Republic and Mexico confirmed their first cases on 24 May and 28 May, respectively. By 2 June, 27 non-endemic countries had reported or identified monkeypox cases [4], all of which were infected with the West African clade. Interestingly, Belgium was the pioneer in implementing a 21-day quarantine period for mpox [20].

The first cases were reported in individuals who had travelled from the endemic regions of Africa to Europe and North America, and then, spread the virus. However, the sudden onset and wide geographic distribution of numerous cases, as well as its emergence in developed countries, suggest widespread person-to-person transmission. If the MPXV becomes established in non-endemic countries as a widespread human pathogen, the risk to public health could increase significantly [20].

As of the end of May 2023, a total of 111 countries around the world had reported 87,545 confirmed monkeypox cases. Out of these cases, 85,962 have been reported in locations that have no historical record of monkeypox outbreaks, while 1583 cases were reported in areas with a history of monkeypox outbreaks, according to the Center for Control of Diseases (Figure 1) [21].

The global scenario in which monkeypox emerged is important, particularly given the ongoing SARS-CoV-2 pandemic and the Ukraine–Russia conflict. These events have raised concerns about the possibility of successive viral pandemics, highlighting the urgent need to enhance public health preparedness for future zoonotic epidemics [22].

## 3. Mpox Cases in the Post-COVID-19 Pandemic Period

After nearly three years of the global health and economic impact caused by the COVID-19 pandemic, it is crucial to broaden our focus and consider the potential impact of other infectious diseases such as monkeypox [23]. Health systems need to be adequately prepared to respond to potential outbreaks of mpox, in order to effectively prevent and control its spread.

Recent cases of mpox have raised public concern, as the majority of cases have been observed in a younger population without a specific vaccine or cross-immunity, and with no apparent travel links to Africa. These factors raise alarming implications for the spread of monkeypox [23]. This underscores the need for continued surveillance and research to better understand the risk factors and transmission dynamics of mpox, as well as to develop effective control and prevention measures, such as a vaccine.

Although mpox typically has a mild clinical course and a low rate of transmission compared to SARS-CoV-2 [24], it can still cause complications in some cases. Approximately 25% of mpox cases present with pulmonary failure, which is a common symptom of both mpox and SARS-CoV-2 infections. These similarities raise concerns for health systems, as both viruses can potentially strain resources and healthcare workers [23].

SARS-CoV-2, the virus responsible for the COVID-19 pandemic, and mpox, which causes monkeypox, have undergone significant evolutionary changes. RNA viruses, such as SARS-CoV-2, are known for their high mutation rates, allowing them to rapidly adapt and spread. In contrast, DNA viruses, such as mpox, have a lower mutation rate and typically acquire only 1–2 mutations per year [25]. However, evidence suggests that mpox has acquired at least 50 mutations compared to previous outbreaks, which may affect its transmissibility [26]. These changes do not necessarily imply a high mutation rate, but rather, may be a product of the duration of time that the virus has been circulating in humans [27]. Therefore, the number of mutations can be considered an indicator of how long the virus has been present in humans. Data suggest that mpox had been present in humans for several years before the alarming cases that emerged in 2022.

In November 2022, El-Qushayri, Reda, and Shah conducted a systematic review of patients co-infected with COVID-19 and mpox. The review identified three cases of co-infection, all of which required hospitalization. However, none of the patients developed severe outcomes [28]. The authors highlighted that the early detection of cases is crucial in reducing the severity of co-infection and controlling the spread of both diseases, particularly in populations at risk.

This similarity in the clinical presentation of both viruses highlights the necessity for continuous monitoring and research to comprehend the potential risks and impacts of mpox, especially in areas where the virus is circulating and among populations at risk. It is essential to develop effective treatment strategies for patients in order to prevent severe outcomes and control the spread of these diseases.

## 4. Animal Reservoirs

Historically, the naming of species within Orthopoxvirus has been based on the host animal from which they were first isolated [29]. Zoonotic Orthopoxviruses are primarily isolated from animals that are incidental hosts and in close proximity to humans, with wild animals serving as the natural carriers. However, the name of a zoonotic Orthopoxvirus species does not necessarily indicate its reservoir animal. As mentioned previously, Orthopoxviruses are remarkable for their wide spectrum of hosts, from humans to wild animals, and can either be host-restricted or have a broad host range. For example, VARV only affects humans, while mpox, CPXV, and VACV are generalist zoonotic Orthopoxviruses [29].

The initial case of monkeypox infection was identified in *Macaca fascicularis* or cynomolgus macaques [2], as well as other non-human primates such as chimpanzees (*Pan troglodytes*) and orangutans (*Pongo pygmaeus*). Monkeypox infections have also been identified in rodents, such as hamsters, porcupines (*Atherurus africanus*), mice (*Mus musculus*), woodchucks (*Marmota monax*), jerboas (*Jaculus* sp.), and rabbits (*Oryctolagus cuniculus*) [29].

According to the susceptibility to infection, monkeypox can further infect a wide range of mammal species, including short-tailed prairie opossums (*Monodelphis domestica*), black-tailed prairie dogs (*Cynomys ludovicianus*), ant-eaters (*Myrmecophaga tridactyla*), African hedgehogs (*Atelerix* sp.), and southern opossums (*Didelphis marsupialis*) [12]. In nature, the virus has been isolated from wild animals, like the sooty mangabey (Cercocebus atys) [30] and rope squirrel (Funisciurus anaerythrus) [31,32]. The origin of the MPXV remains unclear, as the natural reservoir remains unidentified, but some data suggest that African rodents [33] may be responsible for maintaining the virus’ circulation in nature. Black-tailed prairie dogs, which are sold as pets, may also play a role in the spread of the MPXV through animal–human interaction.

## 5. Viral Attachment and Pathogenesis

The poxvirus family includes DNA viruses with genomes ranging from 150,000 to 300,000 base pairs. The monkeypox genome is a linear, double-stranded genome of 197 kb encoding more than 200 proteins related to genome replication, virion assembly, cell entry, and transcription. The MPXV is an enveloped virus with a size of 200–250 nm [34]. It is considered a self-sufficient DNA virus that relies on host ribosomes to translate its viral proteins but can encode its own replication, transcription, and mRNA biogenesis machinery for replication in the cytoplasm of the host cells [35].

As described above, the MPXV can enter the body through various routes, including the oropharynx, the nasopharynx, the intradermal route, and sexual transmission [7,20]. The primary target of the virus is the lymphoid tissue, but its tropism has also been reported in the ovaries, salivary epithelium, brain, heart, kidney, liver, pancreas, and lungs [36,37].

After the virus enters the host cells, early gene transcription events take place at perinuclear sites called viral factories [38]. These factories are derived from a single infecting particle [39] and produce two infectious forms: intracellular mature virions (MV) and extracellular enveloped virions (EV) [35]. MVs are stable in the external environment and have a single outer lipoprotein bilayer that encloses the viral core. They are flanked by protein structures known as lateral bodies, and 5% of the MV mass from other poxviruses is constituted by lipids. This lipid content has been described as a key feature in the cell uptake of the virus by macropinocytosis [37].

EVs are formed from the transport Golgi apparatus or endosomes and have one additional membrane (compared to MVs). They are released by exocytosis [40], facilitating efficient spread from cell to cell and resistance to host immune system mechanisms [37]. MVs and EVs are released during host cell lysis [35].

The invasion of the MPXV involves several steps, including entry, membrane cell fusion, and core invasion [37]. MVs enter host cells by macropinocytosis, while EVs fuse with the plasma membrane at a neutral pH [37]. This process is similar to that of other poxviruses and allows for the virus to enter and infect host cells efficiently. Although specific cell receptors for the MPXV have not been identified, the high homology between monkeypox and vaccinia viruses suggests that they may share some similarities regarding entry (Figure 2) [40].

Mpox and VACV have orthologs E8:D8, A29:A27, A28:A26, and H3:H3, respectively [40]. The adsorption of these viruses on the surfaces of cells is facilitated by H3 and A27, which bind to heparan [37,41]. These two viral proteins are important for the infectivity of VACV. Additionally, A26 binds to laminin [42] and D8 binds to chondroitin [43].

The membrane fusion and core invasion processes are activated by a low pH [41] and require the formation of the entry fusion complex (EFC), which is composed of various viral proteins. In VACV, the EFC includes the viral proteins A16, A21, A28, F9, G3, G9, H2, J5, L1, L5, and O3 [44], while, for the MPXV, all these viral proteins except O3 are necessary for its viral biosynthesis [40].

After viral entry, the virus starts to replicate and encode all the necessary proteins for structural assembly as a self-sufficient DNA virus [45,46]. However, this depends on the host protein synthesis machinery [47]. Then, MVs are prepared for packaging by Golgi membranes and become intracellular enveloped viruses (IEVs). IEVs undergo fusion with the plasma membrane, resulting in the formation of a cell-associated enveloped virus [48], which can either be removed from the cell (extruded cell) or released as an extracellular enveloped virus (EEV). It is also possible for the MVs to bud directly and release the EEV, thereby bypassing the formation of the IEV [49].

The interaction between the virus and host cell’s response has been extensively studied, demonstrating significant genetic diversity between genes encoding Clade I and Clade II. This contributes to the described differences in pathogenicity between them [7]. The monkeypox genomes contain several genes encoding host-response modifier (HRM) proteins, including poxviral inhibitors of complement enzymes (PICEs) and the MOPICE protein, which are considered virulence factors of mpox. The MOPICE protein was described to contribute to the higher pathogenicity of Clade I, as it is not present in Clade II [45].

Poxviruses also encode other virulence factors, including host range factors (Hrfs), which are proteins that modulate the host response to viral infections. For instance, BR-203, an Hrf protein, cell-specifically inhibits the antiviral host response by inhibiting IL-1 and IL-1B receptor binding [40].

Realegeno et al. (2017) described that mpox virus infection in human haploid cells alters Golgi trafficking by affecting vacuolar protein sorting (VPS), including VPS52 and VSP54. They found that *VPS52* knockout reduced the viral spread [46], indicating that these factors contribute to the virulence of mpox. Another factor produced by mpox is the protein A47R, which is known to interact with MyD88, TRIF, and TRAM, adaptor proteins that trigger inflammation and immune responses [50].

After evading the host immune response, mpox is able to spread through the host. During primary viremia, the virus spreads to the local lymph nodes, a feature responsible for lymphadenopathy [51]. Then, mpox can spread through circulation and achieve viral tropism to other organs, which is referred to as secondary viremia [7].

## 6. Immunopathophysiology

The innate immune response is the first line of defense against viral infections, which includes monocytes and neutrophils. However, these immune cells have been shown to be targets of mpox and may contribute to virus dissemination [52]. While the innate immune response is important in controlling local pathogenesis in tissues, it is not effective in preventing the spread of monkeypox infection [53]. The host’s immune response to mpox infection involves the activation of innate immune cells such as natural killer cells and inflammatory cells and the production of interferons (IFNs) and the complement system, which work together to clear the virus and facilitate cell and humoral responses, including antibody production and viral clearance mechanisms. However, despite these immune mechanisms, mpox has developed immune evasion strategies to bypass the host’s defense and establish a pathogenic infection [49].

MPXV has been shown to directly harm immune organs such as the spleen, tonsils, lymph nodes, and thymus, leading to lymphopenia. The complement system is also affected by monkeypox and variola virus infections. This interference is mediated by the secretion of viral proteins, known as virokines [54], which mimic complement regulators and interfere with complement activation. Among these virokines, the VV complement control protein (VCP) has been identified as a key player in impairing the complement system in poxvirus infections [55].

Phagocytosis is a crucial innate immune mechanism, particularly in dendritic cells, which play a key role in activating T cell responses. However, poxviruses have been shown to interfere with this process by using semaphorin family proteins that bind to the receptor Plexin C1. This interaction inhibits phagocytosis in neutrophils and dendritic cells [56], which is involved in the immunopathogenesis of monkeypox [49].

The immune response to viruses relies heavily on the type I IFN response, which is primarily mediated by NK cells. However, the MPXV has developed strategies to evade this response. For instance, the virus inhibits the expression of chemokines such as CCR6, CXCR3, and CCR5, leading to a reduction in the secretion of IFNγ and TNFα [57]. Additionally, the MPXV encodes the F3 protein, which inhibits the activation of PRRs [58] and the phosphorylation of the eukaryotic translation initiation factor 2 (eIF2α) and PKR, thereby reducing IFN production [57]. This impairment of the type I IFN response is a critical factor in the immune evasion of the virus, as it compromises the killing functions of NK cells, including degranulation, migration potential, and T helper 1 (Th1) polarization, which is essential for the adaptative immune response [59].

Regarding cell-mediated immunity, CD4^+^ T cells also play a crucial role in combating viral infections through various mechanisms. They recruit dendritic cells (DCs) and CD8^+^ T cells to the areas of viral replication and also stimulate the differentiation of CD8^+^ T cells into effector and memory T cells. These T cells, in turn, promote the production of virus-specific antibodies by interacting with B cells. However, the MPXV has developed mechanisms to evade antiviral cell responses, including the suppression of T cell activation. Specifically, the MPXV can inhibit T-cell-receptor-mediated T cell activation by using alternative antigen presentation pathways [60].

B cells play a critical role in the immune response against viruses by producing antibodies and generating immunological memory. The effectiveness of this strategy is demonstrated by the success of the global campaign to eradicate smallpox [52]. Orthopoxviruses contain molecules known as immunoevasins that can interfere with viral recognition and clearance, including the MHC class I-like protein (OMCP), which is encoded by mpox and cowpox virus (CPXV) [61]. OMCP functions as a soluble antagonist of the NKG2D receptor, which is expressed by NK and T cells, and also binds the Fc receptor-like 5 (FCRL5) receptor, which is highly expressed by B cells [62].

Mpox infection triggers a complex B cell response that results in the production of monoclonal antibodies against various mpox antigens, such as H3, A27, D8, B5, A33, and L1. These antibodies have been shown to confer protection against systemic infection [63]. Soluble signatures identified in mpox have also been used to describe the antibody profile, with IgM being identified as a biomarker of disease severity [52].

During mpox infection, a cytokine storm similar to that described during SARS-CoV-2 infection can occur, leading to multiple organ failure [52]. Severe mpox infection has also been associated with increased blood tyrosine levels or hypertyrosinemia in cynomolgus monkeys [53]. In humans, hypertyrosinemia has also been correlated with disease severity as a result of a higher Th2 immune response and a decreased Th1 response [64].

The state of dysfunction of the immune response that could be temporary or permanent is defined as immunosuppression, which could be due to disease conditions such as HIV/AIDS, hematologic malignancies, autoimmune illness, innate immunodeficiencies and solid-organ transplantation or induced by medication such corticosteroids [65].

In MPXV infection, immunosuppression compromises both the innate and adaptive immune responses, leading to reduced virus clearance and prolonged viral shedding, diminished Th1 response [66], and unable to mount an antibody response, contributing to the delayed control of viral replication and increase in disease severity [67] with extensive skin rashes [68,69], increased viral, bacterial, and fungal secondary infections as multiple organ failure [65]. 

The scientific perspective explores the concept of immunity conferred by MPXV natural infection and smallpox vaccination provides robust and enduring immunity [70], making reinfections unlikely; however, MPXV infection in immunocompromised patients, such as patients with HIV/AIDS and those undergoing immunosuppressive therapy, have been reported, and the severity can be more pronounced, potentially leading to severe diseases outcomes [71,72,73]. 

Further research is required to elucidate the immunological mechanisms underlying MPXV infection and the impact of viral genetic variation on re-infections in immunocompromised individuals with longer follow-up.

## 7. Modes of Transmission

The MPXV is transmitted to humans through two different routes: from infected animals to humans, also known as primary transmission, and through human-to-human transmission, which is known as secondary transmission [74]. MPXV infections in humans are mainly found in regions of western and central Africa where humans and non-human primates are in close proximity, such as in forest areas. However, the specific animal reservoirs and animal species involved in the first transmission of the MPXV to humans remain unknown [39].

Human infection with the MPXV has been linked to handling, hunting, and bushmeat consumption, non-human primates, and other small mammals [51]. Proximity to rodents has also been recognized as a potential cause of human infection. The destruction of wild animals’ habitats has also been associated with the propagation of the virus to humans [75].

The first known case of monkeypox transmission from an intermediate host to humans outside of Africa was described in the USA in 2003. Here, prairie dogs were identified as the intermediate host, which could have come into contact with the MPXV through infected rodents [28,31]. Animal transmission can occur through different routes, such as bites, the preparation and consumption of bush meat, scratches, or contact with infected bodily fluids [8].

Direct contact with the skin lesions and respiratory secretions of individuals who are infected, as well as contaminated objects, such as clothing, are significant risk factors for monkeypox transmission [14]. Mpox is an infectious disease that can be contracted through the exposure of the oropharynx, respiratory mucosa, broken skin, or mucous membranes of the host [7]. In addition, mpox can be transmitted vertically during pregnancy through the placenta, a process known as congenital monkeypox [20,76].

The transmission rates of infectious diseases vary depending on economic, social, and environmental factors. The current monkeypox outbreak has disproportionately affected younger populations (<50 years), with most cases occurring within the population of gay, bisexual, or men who have sex with men (gbMSM) [77]. Sexual contact has been described as a mode of transmission occurring through genital lesions [75,78]. It has been speculated that the virus may have tropism for the testes, as the virus has been identified in semen [79] and there have been human cases with exclusive peritonsillar lesions [80].

Mathematical models indicated that the reproduction number (R) for mpox was >1 in 2020, indicating epidemic potential [8]. However, the recent outbreak in 2022 in Europe has shown higher estimates of R in males, particularly in gbMSM as the core group. In November 2022, Branda et al. used a mathematical model that included countries with an incidence higher than 10 new daily cases (Belgium, Italy, the United Kingdom, Portugal, the Netherlands, France, Switzerland, Spain, and Germany) and found a median R of 2.44. It should be noted that this higher estimation of R referred to males, mainly gbMSM, and is not applicable to the general population [81].

The modes of mpox transmission previously described suggest that the MPXV can spread through close contact. However, further investigation is needed to fully understand the routes of transmission. It is also important to use animal models to better understand the susceptibility to mpox and the role that different animal hosts may play in the transmission of the virus. This information can be used to develop targeted interventions that are more effective in managing the transmission of zoonotic diseases.

## 8. Laboratory Diagnosis

Rapid diagnosis is crucial in limiting outbreaks of infectious diseases, including mpox. However, clinical observations alone are not sufficient for making a definitive diagnosis. The WHO warns that several other conditions that cause skin rashes, such as bacterial skin infections, scabies, chickenpox, syphilis, medication-associated allergies, measles and Orf [82], which can present clinical symptoms that could be clinically indistinguishable from mpox [83]. Therefore, it is important to use laboratory testing and confirmatory diagnostic procedures to accurately diagnose mpox.

A real-time polymerase chain reaction (RT-PCR) is used as the confirmatory diagnostic test for mpox [84]. Guidelines recommend targeting at least two genes that are conserved across all known circulating MPXV clades, with at least one target that is MPXV-specific. The genes that are targeted to distinguish mpox include the TNF receptor gene [85], the DNA polymerase gene *E9L* [86], the envelope protein gene *B6R* [87], DNA-dependent RNA polymerase subunit 18 (*RP018*) [40], *B7R* [88], *FL3* [88], *N3R* [40], complement binding protein *C3L* [40], the core protein gene *CP* [89], and the open reading frame *O2L* [90].

To diagnose mpox infection, a range of specimen types can be used for laboratory testing. These include skin biopsies of vesiculopustular rash lesions, intact skin vesicular lesions, and exudate or crust from symptomatic individuals in areas with the highest virus concentration [91]. In addition to these samples, mpox DNA has been detected in oropharyngeal, nasopharyngeal, anal, urethral, conjunctival swabs, and semen samples [92]. Other diagnostic modalities include electron microscopy of the rash after staining, the identification of specific antigens through immunohistochemistry, viral isolation, and the culture and detection of IgM/IgG antibodies [35].

Commercial assays such as Tetracore Orthopox BioThrear Alert^®^ are available as rapid-flow-based tests developed for field use [93]. The Antibody Immuno Column for Analytical Processes, or ABICAP, is an immunofiltration tool based on gravity-driven flow-through antigen-capture ELISA, which detects vaccinia, cowpox, monkeypox, and variola viruses [94]. However, serological and protein-detection-based diagnostic tests have cross-reactivity within Orthopoxviruses and do not provide mpox-specific confirmation. Electron microscopy is not highly sensitive, it is expensive, and the time required to analyze the sample is long, which limits its application [40].

Other laboratory findings associated with monkeypox infection have been described in the literature [91]. Common laboratory findings among monkeypox-infected patients include elevated levels of alanine aminotransferase (ALT) and aspartate aminotransferase (AST), leukocytosis, hypoalbuminemia, low blood urea nitrogen levels, and thrombocytopenia [88,89]. These alterations have also been reported as predictors of poor prognosis [95,96].

The diagnosis of mpox is challenging, as confirmatory testing requires samples to be sent to a public health laboratory or one of five authorized commercial labs for analysis, making the diagnostic process less accessible for patients. Additionally, there are currently no options for testing at home or at point-of-care facilities, as are available for COVID-19 diagnosis. As outbreaks of monkeypox evolve and case counts rise, it is becoming increasingly crucial to enhance the ease, accessibility, and diagnostic capabilities of mpox testing to prevent misdiagnosis and improve the treatment of individuals with suspected infections. Therefore, efforts to develop reliable and rapid point-of-care tests for mpox, as well as to improve laboratory infrastructure in areas at risk of outbreaks, should be a priority.

## 9. Clinical Features

In the 2022 outbreak, the average time between exposure to monkeypox and the onset of symptoms was described as being 7.6 days (IQR, 6.5 to 9.9) [97].The interval between the onset of symptoms in a primary case and the onset of symptoms in a secondary case, also known as the serial interval, was reported to be seven days for rash onset [98]. Previously, it was not recognized that transmission could occur during the incubation period [99].

Monkeypox infection typically follows a prodromal phase lasting 1–4 days and is characterized by non-specific symptoms such as general discomfort, chills, fever, muscle pain, back pain, headache, and respiratory symptoms [25]. The progression of infection also includes lymphadenopathy [99], lasting approximately 0–2 days. Next, a vesiculopustular rash develops [100], lasting for 7–21 days [40]. The rash typically initiates on the face and subsequently spreads to involve the oral mucosa, soles of the feet and palms, conjunctiva, and perigenital, perianal, and perioral mucosa, over the course of 2–4 weeks. The development of the rash follows a series of stages, beginning with macules (1–2 days), followed by papules (1–2 days), and then vesicles (1–2 days), pustules (5–7 days), and ultimately, scabs (7–14 days) [100]. During the initial stage of the rash, which lasts for around a week, the infected individual is considered highly infectious [25].

The areas affected by monkeypox skin lesions have been classified as “partial-thickness wounds”. Pathological changes in skin lesions, such as ulceration, tissue death, and excessive growth of cells, become more severe as the pustules progress and can lead to issues such as cellulitis and subsequent bacterial infections if left untreated [39].

Although mpox often clears without treatment [40], the severity and risk of mortality can be influenced by factors such as the clade of the infecting mpox strain [20], the route of exposure [101], and the immune status of the person [52]. The mortality rate for mpox ranges from 1 to 10% [20].

The complications associated with mpox are numerous and can have serious consequences for the patient. Bacterial superinfections, corneal infections, scarring, bronchopneumonia, septic shock, cellulitis, respiratory distress, and encephalitis have all been reported as potential complications of mpox [7]. In addition, retropharyngeal abscesses [96] and dehydration [39] have also been reported as complications. Dehydration is often due to gastrointestinal symptoms, such as vomiting and diarrhea, as well as lesions in the mouth and throat, which can make it difficult for patients to drink and eat [33].

Although these complications are rare, they have been mainly described in unvaccinated individuals who were infected with smallpox (74%) [51].

## 10. Animal Models Employed in the Study of Mpox Infection

Different animal models have been employed to investigate MPXV infection and to evaluate the efficacy of treatment strategies. This has become increasingly important as a preparedness activity against transmissible viruses during the ongoing COVID-19 pandemic [102].

Closely related mammals have been used as experimental models challenged with MPXV infection to simulate the natural mode of transmission. These models have several important characteristics, including infection doses similar to those that cause disease in humans, and morbidity, mortality, and disease courses equivalent to those in humans [103]. The results of these studies are detailed below, and the primary features of each animal model are summarized in Table 1.

To gain insights into the natural history of the MPXV, its potential reservoir host species, its routes of exposure, and the efficacy of antiviral drugs, researchers have utilized various animal models. These models take into account important factors such as genetic diversity, animal age, inoculation route used, viral dosage, MPXV clade, and time to mortality or death (Table 1).

Different routes of inoculation have been tested in experimental monkeypox infection using guinea pigs and golden hamsters, including intranasal, oral, and intracardial routes [104,105]; however, despite the use of different dosages of the virus, these animals did not show signs of disease. In rabbits (chinchilla strain), susceptibility to MPXV infection was found to be affected by age and the route of inoculation. Adult rabbits were found to be resistant when the inoculation route was intradermal or inoculation was performed on scarified skin. However, when the inoculation route was intravenous, they showed symptoms of general disease with fever and rash. In 10–12-day-old rabbits, infection occurred through the airborne-droplet mode, with high lethality [103,104]. However, the intravenous and intraperitoneal routes do not represent the natural transmission routes of the virus [106].

Prairie dogs have been identified as a useful model in studying the progression of the infection [75]. Hutson et al. challenged these animals with the MPXV and found that the Congo Basin clade of MPXV is more transmissible through the respiratory route compared to the West African clade [107].

Several studies have established the mouse as a lethal model of MPXV infection, particularly the STAT1-deficient C57BL/6 mouse. When infected intranasally, this model led to 100% mortality by day 10 of post-infection, making it an ideal murine platform for the investigation of immunological and pathological responses, as well for prophylactics and therapeutics against MPX infection [106].

Another rodent used as a model for MPXV infection is the Gambian pouched rat. The virus can infect these animals, and they can serve as a source of transmission to both humans and other animals [108]. A significant feature of this model is that, even when infected rats exhibit clinical symptoms, they do not become moribund, making them a potentially important source of MPXV transmission to humans [75].

Squirrels have also been investigated for their contribution to the epidemiology of MPXV in central Africa. This animal model has been used to determine tissue tropism, with in vivo bioluminescent imaging being a useful tool [109]. While MPXV infection in these animals did not cause hepatic or splenic damage, they were found to serve as amplifying hosts, shedding a high amount of virus [75].

According to all the results that were reviewed, the respiratory challenge is likely the best route by which to assess the MPXV pathological responses.

**Table 1 pathogens-12-00947-t001:** Animal models used to study pathological and therapeutic responses to MPXV infection.

Animal	MPXV Clade	Inoculation Route	Dosage Used (PFU)	Mortality (%)	Time to Death
**Guinea pigs** [105]	WA	Intravenous, intracerebral, intracardiac, intraperitoneal, intranasal, intradermous, oral, and scarified skin	10^1^–10^2^	0	NA
**Rabbits**					
Adult rabbit [105]	WA	Intravenous	10^7^	8	1-month post-infection
		Oral	10^9^	0	NA
		Intradermal	10^5^	0	NA
10–12 days old [105]	WA	Oral	10^6^	85	4–14 days
		Intranasal	10^6^	83	4–5 days
**White rats** [105]					
Adult	WA	Intravenous, intranasal, cutaneous	10^1^–10^3^	0	0
1–3-day-old white rats	WA	Intranasal	10^1^–10^3^	100	5–6 days
**White mice** [105]					
8–15 days old	WA	Intranasal, intraperitoneal	1.2 × 10^6^	100	Not provided
		Oral	1.2 × 10^6^	40	Not provided
		Foot pad	6 × 10^2^	100	Not provided
		Intradermic	1.2 × 10^6^	50	Not provided
12 days old	WA	Oral	1.2 × 10^6^	14	Not provided
15 days old	WA	Intranasal	1.2 × 10^6^	100	Not provided
**Hamsters** [105]					
	WA	Intranasal, oral, intracardiac, scarified skin	1.5–5.9 × 10^7^	0	NA
**Squirrels**					
Ground squirrels [110]	WA [111]	Intraperitoneal Intranasal	10^5.1^ 10^6.1^	100	9 days
Rope squirrels [109]	CB	Intranasal Intradermal	10^6^	75 50	13 days 11 days
**Mouse**					
C57BL/6 lab mice [106]	CB	Intraperitoneal Intranasal	5 × 10^4^	0	
CAST/EiJ [112]	CB	Intranasal	10^4^–10^6^	100	5–8 days
			10^3^	60	
			10^2^	0	NA
	WB	Intranasal	10^5^–10^6^	100	8 days
			10^4^	50	
			10^3^	12.5	
			10^2^	0	NA
SCID [106]	CB	Intranasal	275	100	16.8 days
DBA A/Ncr C3HeJ [106]	CB	Intranasal Footpad injection	5 × 10^4^	0	NA
BALB/c IFN-yR−/− [106]	CB	Intranasal	990	0	NA
Type 1 IFNR−/− [106]	CB	FP	6 × 10^3^	0	NA
C57BL/6 stat1−/− [106]	CB	Intranasal	470	90	9.3 days
129 stat1−/− [106]	CB	Intranasal	4700	40	10 days
SCID-BALB/c mice [113]	WA CB	Intraperitoneal	10^5^	100	9 days
Dormouse (*Graphiurus kelleni*) [114]	CB WA	Footpad injection	10^4^	92	7–10 days
	CB	Intranasal	2000–200	100	7.9–8.7 days
Gambian pouched rat					
[115]	WA CB	Scarification	10^4^	25 0	13 days NA
[108]	CB	Intradermal Intranasal	10^6^	0	NA
**Prairie dogs** (*Cynomys ludovicianus*)					
[116]	CB	Intranasal	10^4.5^	25	13 days
	CB	Intradermic	10^4.5^	50	11–12 days
	WA	Intranasal Intradermic	10^4.5^	0	NA
[117]	WA	Intraperitoneal	10^5.1^	100	8–11 (IP route)
		Intranasal	10^5.1^	0	NA
[118]	WA	Intranasal	10^4^	33	14 days
[119]	CB	Intranasal	10^6^	0	NA

CB: Congo Basin region; IP: intraperitoneal; NA: non-applicable; PFU: pock-forming units; WA: West Africa.

## 11. Cross-Immunity and Vaccines

Orthopoxviruses are a group of viruses that share common genetic and antigenic characteristics. It has been described that infection with any of these species can provide significant protection against other species [63,120]. For example, the vaccinia virus vaccine can protect against diseases caused by other Orthopoxvirus species, such as VARV, mpox, or CPXV. Similarly, the smallpox vaccine can confer cross-protection against mpox in humans [121,122]. Furthermore, pharmacological treatments that were originally developed for smallpox were found to be effective against mpox infection, as demonstrated during the 2022 outbreak [123].

The cessation of smallpox vaccinations in 1978 led to a decrease in the cross-protective immunity against different Orthopoxviruses [124], especially among younger individuals who have not received the vaccinia virus vaccine. As a result, the global population of susceptible individuals has been increasing, leading to a rise in the frequency and geographical distribution of human mpox in recent years [91].

Poxviruses are closely related genetically, which enables cross-reactive immune responses between them through memory cells and antibodies that can recognize common viral antigens [49]. To prevent and control mpox, the WHO has recommended vaccination campaigns using the smallpox vaccine, particularly among high-risk populations and healthcare workers [84,124].

Dryvax^®^ (Wyeth Laboratories, Inc.) is a first-generation smallpox vaccine composed of a pool of vaccinia virus strains that vary in their degrees of virulence. It is a live virus preparation of vaccinia virus closely related to the smallpox virus. However, it can have life-threatening adverse effects, including myopericarditis, and poses a risk to pregnant and immunocompromised individuals [125]. Due to the successful eradication of smallpox, routine vaccination with Dryvax is no longer recommended [100].

The second-generation vaccine ACAM2000^TM^ is a smallpox vaccine that was developed during the eradication campaign. It is a purified clone isolated from Dryvax and was approved by the FDA in 2007 for use in high-risk individuals, such as laboratory workers and military personnel; however, it is contraindicated in individuals with immunodeficiency [99] and is listed as pregnancy category D. During the 2003 outbreak in the USA, ACAM2000 was demonstrated to be effective in reducing the symptoms in infected individuals [100].

JYNNEOS, also known as Imvamune or Imvanex, is an alternative to ACAM2000. It is a replication-deficient modified vaccinia virus vaccine that was licensed by the United States for smallpox and monkeypox in 2019 [120]. In 2021, the Advisory Committee on Immunization Practices (ACIP) recommended JYNNEOS for individuals who had been in contact with Orthopoxviruses [126]. The vaccine is administered in a two-dose regimen and can also be used as a booster for those who received ACAM200 as their primary vaccination [35]. Therefore, two vaccines are now available as preexposure prophylaxis against Orthopoxvirus infection [126].

Another experimental vaccine was developed to prevent smallpox outbreaks, known as the Aventis Pasteur Smallpox Vaccine. This vaccine was developed from a replication-competent vaccinia virus, similar to the ACAM2000 vaccine. However, it is only used in the USA under circumstances where ACAM2000 and JYNNEOS are not available [127].

In 2022, and during the COVID-19 pandemic, Mucker et al. developed a vaccine against monkeypox known as the 4pox vaccine. This vaccine targets the L1, A27, B5, and A34 Orthopoxvirus proteins and is a DNA-based vaccine that does not require adjuvants or formulations. The vaccine can be delivered via electroporation or the intramuscular route. The protective efficacy of the 4pox vaccine was evaluated using an animal model of respiratory infection and low doses associated with human smallpox exposure. These results demonstrated that the intramuscular route of administration can protect against aerosol exposure and elicit neutralizing antibodies (Table 1) [128].

The development of vaccines against monkeypox could be accelerated through the use of bioinformatic analysis. Online resources such as EPIPOX (available at: http://imed.med.ucm.es/epipox/; accessed on 22 April 2023) compile immunoinformatic characterizations of T cell epitopes between Orthopoxviruses [129].

To contain the spread of monkeypox and minimize the risk of infection, a strategy known as “ring vaccination” was implemented among healthcare workers. This approach involves administering vaccines to individuals who have been in close proximity to those infected with the MPXV, with the aim of preventing any further transmission [98,99].

The administration of vaccines, including those for influenza and COVID-19 during the pandemic, has raised concerns [130]. However, to prevent the occurrence of new pandemics, it is essential to establish scientific and clinical protocols based on the lessons learned from the COVID-19 outbreak. Therefore, it is crucial to promptly develop new vaccines to address the potential emergence of unpredictable infections [131].

Encouraging widespread vaccination can be challenging, partly due to the potential occurrence of infrequent, severe side effects associated with available vaccines. Therefore, preventive health measures, including the avoidance of exposure to infected humans or animals and maintaining good hand hygiene, remain the superior approach to disease prevention. The MPXV can be inactivated by heat—specifically, 30 min of treatment at 56 °C—or by chloroform, formaldehyde, sodium dodecyl sulfonate (SDS), and methanol [40].

## 12. Treatment and Prevention

The clinical management of monkeypox is not clearly established, and there are currently no specific treatments approved by regulatory agencies such as the US FDA, WHO, CDC, or European Medicines Agency (EMA) (Table 2). However, some antiviral drugs used for smallpox treatment have been adapted by the FDA, such as brincidofovir and tecovirimat, which were approved in 2018 and 2021, respectively [25]. It is important to note that tecovirimat has not been approved by the EMA [52]. Despite the availability of these drugs, the clinical management of monkeypox infection still involves palliative care, and most patients recover without specific treatment [131,132].

The Strategic National Stockpile through the CDC recommends the use of tecovirimat, cidofivir, and vaccine immune globulin (VIG) for the treatment of poxvirus infections in outbreak situations [120].

Tecovirimat, also known as ST-246 or TPOXX^®^, is an antiviral drug that targets the *F13L* gene and VP37 membrane protein to disrupt viral spread [133] (Figure 2). However, its effectiveness is limited if it is administered more than five days after infection [134].

Cidofivir inhibits the DNA polymerase of poxviruses (Figure 2) and can prevent death when administrated before the onset of the rash, making it a potential treatment option for high-risk contacts and early confirmed cases of monkeypox. However, its use is associated with dose-dependent acute renal failure [122].

Brincidofovir, also known as CMX001 or hexadecyloxypropyl-cidofivir, inhibits viral DNA synthesis by inhibiting DNA polymerase (Figure 2) [102]. In 2021, it received approval from the FDA for smallpox treatment [123], and its efficacy was tested in animal models (Table 2). However, its use in humans (200 mg once a week, oral route) has been shown to result in elevated liver enzymes [135], particularly alanine aminotransferase (ALT), and to lead to an increased risk of mortality in prolonged treatments [123].

VIG is a treatment containing pooled polyclonal immunoglobulins from the plasma of healthy donors, and has demonstrated cross-neutralizing activity against monkeypox in rhesus macaques [136]. The antibodies can be bound to virions, which hinders the virus from infecting new cells (Figure 2) [7]. In 2005, the intravenous formulation (IVIG) received approval from the FDA for the treatment of adverse effects from the vaccinia virus vaccine [137] and has recently been used for mpox infection during the 2022 outbreak [120].

To address the potential development of antiviral resistance, researchers have investigated alternative agents for the treatment of mpox, such as PAV-164 [138] and resveratrol [139], which have demonstrated in vitro efficacy in reducing virus replication (Figure 2) (Table 1). Additionally, other therapeutic approaches have been explored, including immunomodulators, monoclonal antibodies, and cell-based therapy (Table 2).

In general, mpox infection causes a disease that is typically mild to moderate in severity, with a self-limiting course. However, antiviral treatments should be taken into consideration in cases of severe illness that require hospitalization, as well as cases involving undetermined signs and symptoms. It is particularly important to provide antiviral medication to patients at higher risk of severe disease, such as children under eight years old, pregnant women, and immunocompromised individuals.

**Table 2 pathogens-12-00947-t002:** Therapeutic approaches and models for MPXV infection.

Model	MPXV Clade	Therapeutic Agent	Inoculation Route	Dosage Used (PFU)	Mortality (%)	Results
**Animal models**						
Prairie dogs [140]	CB	IMVAMUNE^®^ ACAM2000^®^	Intranasal	10^4^ 10^6^	75 100	Vaccination provided some level of protection to the animals against a challenge of 2 × LD50, but it did not protect them against a challenge of 170 × LD5
Prairie dogs (Cynomys ludovicianus) [141]	CB	ST-246 Tecovirimat	Intranasal	10^5^	0	All animals that were administered ST-246 survived the challenge, and those that received treatment prior to the onset of remained mainly without symptoms
Cynomolgus monkeys (*Macaca fascicularis*) [142]	WA	ST-246 Tecovirimat	Intravenous	5 × 10^7^	0	Administering an oral dose of around 3 mg/kg/day (36 mg/m^2^) to nonfasted NHPs for 14 days, starting at 3 dpi, resulted in complete protection against mortality
Dormouse *(Graphiurus kelleni)* [114]	CB	Cidofovir	Intranasal	75. 4 × 10^3^ 5 × 10^3^	0	Dormice that received a single dose of cidofovir 4 h after being exposed to MPXV showed significant protection against mortality, whereas the group treated with the control presented uniform mortality
	CB	Dryvax vaccine	Intranasal	2 × 10^4^	19	Animals vaccinated with the Dryvax vaccine 4 weeks prior to challenge presented solid protection from mortality when challenged with 2 × 10^4^ PFU of MPXV-ZAI-79
Rhesus macaques (*Macaca mulatta*) [136]	CB	VIG	Intravenous	5 × 10^7^	0	VIG showed promising cross-neutralizing activity sufficient to protect rhesus macaques from a lethal mpox infection
C57BL/6 stat1−/− mice [106]	CB	Dryvax vaccine	Intranasal	4.2 × 10^4^	90	Mice that received a single vaccination of MVA on day-56 or a double vaccination of MVA with a booster on both day-56 and day-28 had comparable survival rates of approximately 90%
	CB	CMX001	Intranasal	5000	0	After being challenged with MPXV, all mice survived, experienced minimal weight loss, and seroconverted. However, when the mice were rechallenged at day 38 after the initial infection, 20% of them died by 8 days post-rechallenge
Marmot model [143]	CB	NIOCH-14	Intranasal	3.4 log10	60	The mechanism of the antiviral action of these compounds is focused on the inhibition of the formation of different enveloped forms of the virus (intracellular, cell-associated, and extracellular)
Rabbits (*Oryctolagus cuniculus*) [128]	RPXV	4pox vaccine	Intramuscular Electroporation	1 × 10^5^	0	The 4pox DNA vaccine protected NHPs, mice, and, in this study, rabbits against fatal infection by MPXV, VACV, and RPXV, respectively
**In vitro**						
HeLa and BSC-40 cells [138]	WA	PAV-164	In vitro	Not provided	NA	At non-cytotoxic concentrations, the compounds demonstrated strong virucidal activity and inhibited infection with VACV, monkeypox, cowpox, and Akhmeta virus when administered before, during, or after viral adsorption
HeLa and human foreskin fibroblasts (HFFs) [139]	WA CB	Resveratrol	In vitro	2 × 10^9^ [144]	NA	Resveratrol reduced VACV and mpox replication. The suppression appears to affect the viral DNA synthesis step
Vero 76, Vero E6 [145]	CB	Ribavirin	In vitro	1 × 10^5^	NA	Ribavirin showed an antiviral function in Vero cells under mpox infection measured by a neutral red uptake assay
HeLa cells, VA (R645), VA-9, and VN36 cell lines [146]	CB	Recombinant IFN-β	In vitro	3.42 × 10^6^ 8.37 × 10^6^ 3.53 × 10^7^ [147]	NA	The induction of the antiviral protein MxA was observed in infected cells treated with IFN-β, and it was demonstrated that constitutive expression of MxA inhibits mpox infection

CB: Congo Basin; HFF: human foreskin fibroblast; NHP: non-human primate; RPXV: rabbitpox virus; WA: Western Africa.

## 13. Concluding Remarks

This review provides a comprehensive overview of the MPXV outbreak, which is currently the seventh public health emergency of international concern (PHEIC), as declared by the WHO in 2022. The emergence of monkeypox highlights the potential risk of viral infections spreading from zoonotic reservoirs. Monitoring monkeypox cases and the viral evolution of the MPXV is essential for identifying potential animal hosts of Orthopoxviruses in Africa and improving diagnostic methods. This paper discusses the transmission, pathogenesis, diagnosis, animal models, new vaccines, treatments, and prevention options for MPXV infection. However, further research, treatments, and vaccines are required to ensure global preparedness for potential monkeypox pandemics and effectively prevent and manage the disease. The current monkeypox outbreak is significant, as it overlaps with the COVID-19 pandemic, which presents challenges for effective disease control and surveillance. The decline in immunity to smallpox is believed to be a factor contributing to the rising prevalence of monkeypox infection in endemic regions. Moreover, the high number of confirmed cases in non-endemic countries, along with cases of human-to-human transmission, highlight the potential for the global spread of monkeypox, emphasizing the need for ongoing surveillance and effective prevention measures. The recent use of the ring vaccination strategy in the 2022 monkeypox outbreak, which was previously used against Ebola, facilitated the efficient control of potential direct cases, highlighting the importance of adapting and applying effective strategies from previous outbreaks to new emerging diseases. However, to prevent and control the future spread of the MPXV, it is crucial to conduct further research, including the evaluation of the effectiveness and feasibility of ring vaccination in different contexts, as well as the development of new antivirals, vaccines, and preventive strategies.

## Figures and Tables

**Figure 1 pathogens-12-00947-f001:**
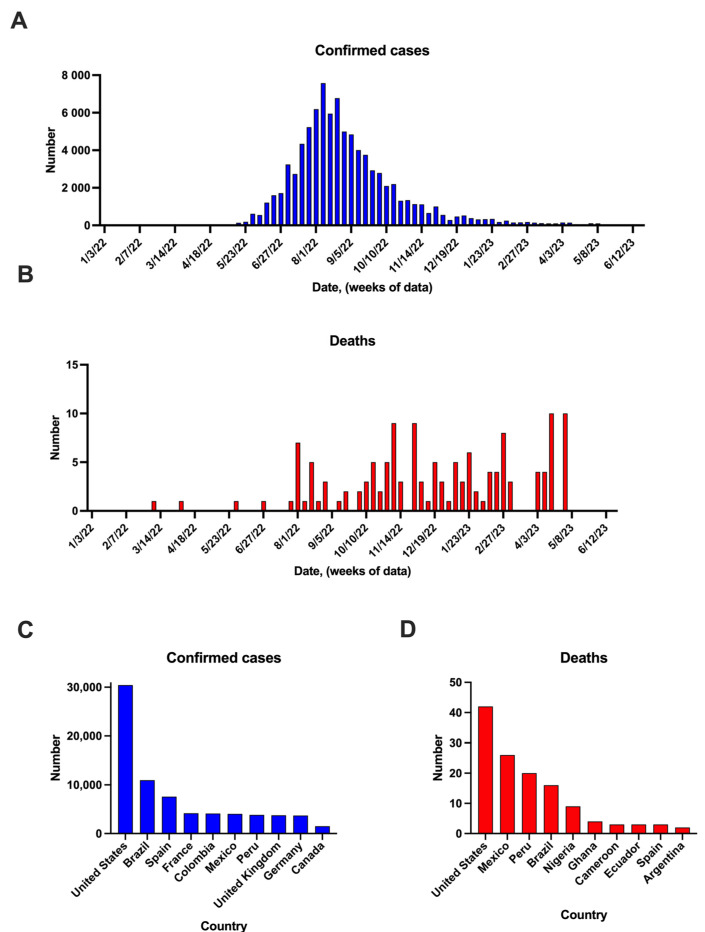
Confirmed mpox cases and deaths in the world (**A**,**B**). Global cases and deaths, respectively (data represent the cases per week) (**C**). Top 10 countries with confirmed cases and (**D**) top 10 countries with reported deaths. Data are derived from the WHO (accessed on 29 May 2023).

**Figure 2 pathogens-12-00947-f002:**
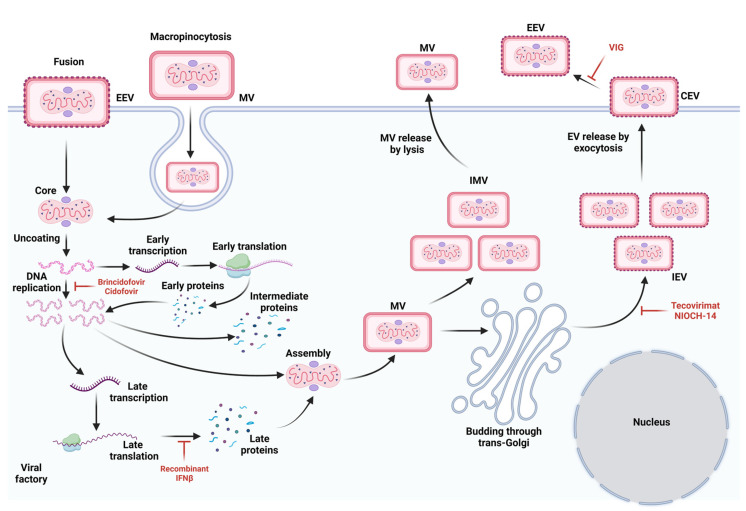
Lifecycle of MPXV. Schematic overview of the processes involved in the attachment, replication, and creation of viral offspring in MPXV is provided, along with the mechanisms of action of therapeutic agents that have been suggested for the treatment of mpox infection. Figure created with BioRender.com (accessed on 3 May 2023).

## Data Availability

Not applicable.

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
