# Peer review of "Human Monkeypox: A Comprehensive Overview of Epidemiology, Pathogenesis, Diagnosis, Treatment, and Prevention Strategies"

_pathogens, 2023, doi:10.3390/pathogens12070947_

Round 1

Author Response

Reviewer #1 (Remarks to the Author):

General Comments

Thank you for the opportunity to review your interesting manuscript. I enjoyed reading it as it is really an updated and comprehensive literature review regarding the Human Monkeypox 2022 Outbreak. Some aspects should be improved in regards of immunopathogenesis and differential diagnosis (see Major Compulsory Revisions). Lastly, there are formal errors which should be corrected throughout the text (see Minor Essential Revisions). English syntax is optimal.

We thank the reviewer for this comment.

 - Major Compulsory Revisions:

Modes of transmission, line 28: Please, try to use a more inclusive language in according to the most recent guidelines (see UNAIDS https://www.unaids.org/sites/default/files/media_asset/2015_terminology_guidelines_en.pdf). “Men who have sex with men” should be written out in full. Abbreviations can be used where and when brevity is required. As an alternative, use the expression gbMSM (gay, bisex or men who have sex with men) as suggested by the WHO (https://www.who.int/news/item/25-05-2022-monkeypox--public-health-advice-for-gay--bisexual-and-other-men-who-have-sex-with-men).

We thank the reviewer for this suggestion. Following the reviewer’s recommendations, we edited this sentence and now it is: “The current monkeypox outbreak has disproportionately affected younger populations (<50 years), with most cases occurring within the population of gay, bisex or men who have sex with men (gbMSM)”. (Page 9, lines 547-549).

Immuno-pathophysiology: Please, expand the discussion by deepening MPXV infection in immunocompromised patients and by analyzing the main aspects of MPXV re-infections. Here, some recently published work that might be included to make the review even more updated: Rocha SQ, et al. AIDS Res Hum Retroviruses. 2023 and Álvarez-López P, et al. Int J STD AIDS. 2023.

Following the reviewer’s comment, this is reflected in the discussion (page 8, lines 453-471) in the new manuscript.

Laboratory diagnosis, lines 3-6: A recently published systematic review (Rossi L, et al. Microorganisms. 2023, https://doi.org/10.3390/microorganisms11051138) stated that MPXV lesions might also resemble Orf virus lesions. Please include this underestimated pathogen in the differential diagnosis of MPXV.

We thank the reviewer for this comment and following this suggestion, we have added this relevant study (page 9, lines 571-573).

- Minor Essential Revisions:

Introduction, line 1: Here you write “monkeypox virus (MPXV)”, then both MPXV and “monkeypox virus” randomly appear throughout the manuscript. Please, replace all the “monkeypox virus” with MPXV throughout the manuscript.

We appreciate this general observation. These changes are reflected in the new version of the manuscript.

Table 1: Table 1 should not be placed in the “Viral Attachment and Pathogenesis” chapter.

We appreciate this comment. We placed this table 1 in the Treatment and Prevention chapter, now table 2. This information is reflected in the new version of the manuscript (pages 15 and 16).

Reviewer 2 Report

The article is well written and the authors provide a complete review about the mpox outbreak 2022. However, some alterations should be made, such as:

- all article: Please standardize the name of the disease to mpox (Monkeypox) and the name of the virus to mpox virus (Monkeypox virus or MPXV) designation in the entire article.

- Abstract: “...presents clinical symptoms similar to those of smallpox, such as lymphadenopathy, fever and vesicular–pustular skin lesions.” -> Lymphadenopathy in absent in smallpox, please change the sentence.

- 1. Introduction: "Lymphadenopathy is a hallmark clinical manifestation of mpox, a distinctive feature frequently used to differentiate it from other poxviruses such as chickenpox, measles and smallpox." -> Chickenpox and measles are not poxviruses, please change the sentence.

- 2. Public Health Importance: "The recent re-emergence of monkeypox was first reported on 6 May 2022, in the United Kingdom, followed by Canada, Spain, Portugal, and the USA on 18 May." -> Should consider changing this affirmation because the first case was reported on 7 May in the United Kingdom. Portugal was the second country to notified mpox cases on 17 May, followed by other countries the next day (https://worldhealthorg.shinyapps.io/mpx_global/ ).

- 12.Treatment and Prevention: “Tecovirimat, also known as ST-24 or TPOXX®, is an antiviral drug that targets the F13L gene and VP37 membrane protein to disrupt viral spread [143] (Figure 2). However, its effectiveness is limited if it is administered more than five days after infection [144].” – change the name ST-24 to ST-246.

- Should consider changing the location of table 1 despite being mentioned in figure 1, it is displaced from the topic where it is located.

Minor editing of English language required.

Author Response

Reviewer #2 (Remarks to the Author):

The article is well written and the authors provide a complete review about the mpox outbreak 2022.

We thank the reviewer for this comment.

However, some alterations should be made, such as:

- all article: Please standardize the name of the disease to mpox (Monkeypox) and the name of the virus to mpox virus (Monkeypox virus or MPXV) designation in the entire article.

We appreciate this general observation. These changes are reflected in the new version of the manuscript.

- Abstract: “...presents clinical symptoms similar to those of smallpox, such as lymphadenopathy, fever and vesicular–pustular skin lesions.” -> Lymphadenopathy in absent in smallpox, please change the sentence.

We thank the reviewer for this observation. We edited this sentence and now it is: “Monkeypox virus (MPXV) is an emerging zoonotic virus that belongs to the Orthopoxvirus genus and presents clinical symptoms similar to those of smallpox, such as fever and vesicular–pustular skin lesions. However, the differential diagnosis between smallpox and monkeypox is that smallpox does not cause lymphadenopathy but monkeypox generate swelling in the lymph nodes” (page 1, lines 17-21).

- 1. Introduction: "Lymphadenopathy is a hallmark clinical manifestation of mpox, a distinctive feature frequently used to differentiate it from other poxviruses such as chickenpox, measles and smallpox." -> Chickenpox and measles are not poxviruses, please change the sentence.

We appreciate this general observation. This change is reflected in the new version of the manuscript and now it is: Lymphadenopathy is a hallmark clinical manifestation of mpox, a distinctive feature frequently used to differentiate it from other contagious disease such as chickenpox, measles, and smallpox (page 2, lines 92-94).

- 2. Public Health Importance: "The recent re-emergence of monkeypox was first reported on 6 May 2022, in the United Kingdom, followed by Canada, Spain, Portugal, and the USA on 18 May." -> Should consider changing this affirmation because the first case was reported on 7 May in the United Kingdom. Portugal was the second country to notified mpox cases on 17 May, followed by other countries the next day (https://worldhealthorg.shinyapps.io/mpx_global/).

We appreciate this general observation. This change is reflected in the new version of the manuscript (page 2, lines 116-117).

- 12. Treatment and Prevention: “Tecovirimat, also known as ST-24 or TPOXX®, is an antiviral drug that targets the F13L gene and VP37 membrane protein to disrupt viral spread [143] (Figure 2). However, its effectiveness is limited if it is administered more than five days after infection [144].” – change the name ST-24 to ST-246.

We thank the reviewer for this observation, we have changed it (page 15, line 1183).

- Should consider changing the location of table 1 despite being mentioned in figure 1, it is displaced from the topic where it is located.

We appreciate this comment. We placed this table 1 in the Treatment and Prevention chapter, now table 2. This information is reflected in the new version of the manuscript (pages 15 and 16).

Reviewer 3 Report

I read the manuscript titled “Human Monkeypox: A Comprehensive Overview of the 2022 Outbreak” by Diana Emilia Martínez-Fernández et al. with great interest. The manuscript sums up much of the state of the art on monkeypox and mpox. In fact, relatively little space is dedicated to the 2022 outbreak. Instead, the authors go over the genetics, immunology, virology, biochemistry, etc of monkeypox. 

I propose the manuscript be completely restructured and resubmitted under a different title, representative of all contents of the manuscript.  I also propose that the authors employ language services to improve the text.

A few comments are in order: 

1) The sections of the manuscript are, in order: Introduction, Public Health Importance, Animal Reservoirs, Modes of Transmission, Viral Attachment and Pathogenesis, Immunopathophysiology, Laboratory Diagnosis, Clinical Features, Mpox Cases in the Post-COVID-19 Pandemic Period, Animal Models Employed in the Study of Mpox Infection, Cross-Immunity and Vaccines, Treatment and Prevention, Concluding Remarks. The order of these sections needs to be reconsidered to ease the presentation of the material. Perhaps, the section Public Health Importance may come just before Concluding Remarks. Modes of Transmission should also come much later in the manuscript. Mpox Cases in the Post-COVID-19 Pandemic Period should follow after Public Health Importance.

2) The flow of the manuscript is awkward and the authors end repeating facts several times in the manuscript; e.g. the death rate associated with mpox, the genus of monkeypox, that 1st mpox human case was a child in DRC, etc.  These repetitions should be avoided.  I believe a better section structure of the manuscript will help.

3) Figure 1C says very little and should be removed. Instead, the authors could present the mpox epidemic data chronologically. 

4) The text needs perusal for English.  I list a few problems.  Abstract: “MPXV has become the

most prominent poxvirus…”  Modes of Transmission: “consumption of meat obtained from rodents…” Clinical Features: “mpox is a self-limiting disease…”  What is a self-limiting disease? 

5) Animal Reservoirs: “In nature, the virus has only been isolated from wild animals twice, once in the sooty mangabey (Cercocebus atys) [27] and once in a rope squirrel (Funisciurus anaerythrus) [28].” Actually, the virus has been isolated more than twice from wild animals.

The manuscript requires English language services.

Author Response

Reviewer #3 (Remarks to the Author):

Comments and Suggestions for Authors

I read the manuscript titled “Human Monkeypox: A Comprehensive Overview of the 2022 Outbreak” by Diana Emilia Martínez-Fernández et al. with great interest. The manuscript sums up much of the state of the art on monkeypox and mpox. In fact, relatively little space is dedicated to the 2022 outbreak. Instead, the authors go over the genetics, immunology, virology, biochemistry, etc of monkeypox. 

I propose the manuscript be completely restructured and resubmitted under a different title, representative of all contents of the manuscript. 

We thank the reviewer for this suggestion and appreciate his valuable feedback on our manuscript.

The reviewer mentioned his concern regarding the title of our manuscript and proposed it be completely restructured and resubmitted under a different title. We understand the viewpoint that the title may seem slightly misleading as we have dedicated considerable space to discussing the broader aspects of monkeypox.

However, the central aim of our manuscript was to provide a comprehensive examination of the 2022 outbreak within the broader context of monkeypox as a disease. The extensive discussions on the multifaceted aspects of monkeypox were intended to give readers a more profound understanding of this disease and to underscore the significance of the 2022 outbreak. This contextualization allows us to delve into the complexities and subtleties of the 2022 outbreak, which might otherwise have been lost.

In light of this, we firmly believe the title aptly represents the content of the manuscript. However, in an effort to address the reviewer’s concern, we revised the abstract and introduction of the manuscript and restructure the order of the sections to more accurately reflect this aim and clarify the broader scope of our work.

I also propose that the authors employ language services to improve the text.

We appreciate this general observation, actually our manuscript has undergone English language editing by MDPI. The text has been checked for correct use of grammar and common technical terms, and edited to a level suitable for reporting research in a scholarly journal. The language services used in this manuscript was completed by an editor with knowledge of our field.

A few comments are in order: 

1) The sections of the manuscript are, in order: Introduction, Public Health Importance, Animal Reservoirs, Modes of Transmission, Viral Attachment and Pathogenesis, Immunopathophysiology, Laboratory Diagnosis, Clinical Features, Mpox Cases in the Post-COVID-19 Pandemic Period, Animal Models Employed in the Study of Mpox Infection, Cross-Immunity and Vaccines, Treatment and Prevention, Concluding Remarks. The order of these sections needs to be reconsidered to ease the presentation of the material. Perhaps, the section Public Health Importance may come just before Concluding Remarks. Modes of Transmission should also come much later in the manuscript. Mpox Cases in the Post-COVID-19 Pandemic Period should follow after Public Health Importance.

We would like to thank the reviewer for the highly positive and helpful comments to improve the manuscript. These changes are reflected in the new version of the manuscript. Now the section order is:

  1. Introduction
  2. Public Health Importance
  3. Mpox Cases in the Post-COVID-19 Pandemic Period
  4. Animal Reservoirs
  5. Viral Attachment and Pathogenesis
  6. Immunopathophysiology
  7. Modes of Transmission
  8. Laboratory Diagnosis
  9. Clinical Features
  10. Animal Models Employed in the Study of Mpox Infection
  11. Cross-Immunity and Vaccines
  12. Treatment and Prevention
  13. Concluding Remarks

2) The flow of the manuscript is awkward and the authors end repeating facts several times in the manuscript; e.g. the death rate associated with mpox, the genus of monkeypox, that 1st mpox human case was a child in DRC, etc.  These repetitions should be avoided.  I believe a better section structure of the manuscript will help.

We thank the reviewer for this comment and following this suggestion, we have avoided these repetitions in the new version of the manuscript.

3) Figure 1C says very little and should be removed. Instead, the authors could present the mpox epidemic data chronologically. 

 We thank the reviewer for this suggestion. Following the reviewer’s recommendations, we removed Figure 1C and we included a new figure 1 (A-D) including mpox epidemic data chronologically.

4) The text needs perusal for English.  I list a few problems.  Abstract: “MPXV has become the

most prominent poxvirus…”  Modes of Transmission: “consumption of meat obtained from rodents…” Clinical Features: “mpox is a self-limiting disease…”  What is a self-limiting disease? 

We thank the reviewer for this comment and following this suggestion, we have rephrased these confusing expressions in the new version of the manuscript.

5) Animal Reservoirs: “In nature, the virus has only been isolated from wild animals twice, once in the sooty mangabey (Cercocebus atys) [27] and once in a rope squirrel (Funisciurus anaerythrus) [28].” Actually, the virus has been isolated more than twice from wild animals.

Following the reviewer’s suggestion, we have edited this sentence (page 5, lines 306-308).

Round 2

Reviewer 1 Report

It is write well and it describe most interesting finding about monkeypox

Author Response

Reviewer #1 (Remarks to the Author):

It is write well and it describe most interesting finding about monkeypox.

We thank the reviewer for this comment.

Reviewer 3 Report

I read revision of the manuscript titled “Human Monkeypox: A Comprehensive Overview of the 2022 Outbreak” by Diana Emilia Martínez-Fernández et al. with great interest. Overall, in the light of my previous comments, I found the revision to be unsatisfactory.

The manuscript has 17 pages. However, the narrative about the 2022 mpox outbreak spans about 3 pages (mostly sections Public Health Importance and Mpox Cases in the Post-COVID-19 Pandemic Period). In conclusion, much of the material in the manuscript is just not represented in the title or the abstract. 

Furthermore, I still find the revised order of the manuscript sections inappropriate. I don’t see how the discussion about the MPXV reservoir follows naturally after the section Mpox Cases in the Post-COVID-19 Pandemic Period. Also, I don’t see how Animal Models Employed in the Study of Mpox Infection follows naturally after Clinical Features. 

The text still needs perusal for English.  I list a few problems. 

1) Introduction, line 36: “Human monkeypox is a zoonotic infection…”  The authors refer to mpox.  However, the term “mpox” is first mentioned only in line 58, without explanation. In line 59, the origins of the ten “mpox” are clarified. This clearly needs to be corrected.

2) Caption of Fig 1, line 108:  “Epidemiological curves of mpox in the world. (A and B) Global cases and deaths reported by epidemiological weeks, respectively…” What are epidemiological curves? What are epidemiological weeks?

3) line 451: “Although mpox is usually resolved spontaneously, recovering without treatment in a few weeks…” The patient recovers, not the disease. I guess the authors wish to convey that mpox often clears without treatment. 

Overall, the revision made by the authors is unsatisfactory.

The manuscript still requires small corrections for the English language.

Author Response

Reviewer #3 (Remarks to the Author):

I read revision of the manuscript titled “Human Monkeypox: A Comprehensive Overview of the 2022 Outbreak” by Diana Emilia Martínez-Fernández et al. with great interest. Overall, in the light of my previous comments, I found the revision to be unsatisfactory.

The manuscript has 17 pages. However, the narrative about the 2022 mpox outbreak spans about 3 pages (mostly sections Public Health Importance and Mpox Cases in the Post-COVID-19 Pandemic Period). In conclusion, much of the material in the manuscript is just not represented in the title or the abstract. 

We thank the reviewer for this suggestion. We have changed the titled to: “Human Monkeypox: A comprehensive overview of epidemiology, pathogenesis, diagnosis, treatment and prevention strategies” (page 1, lines 2 and 3).

Furthermore, I still find the revised order of the manuscript sections inappropriate. I don’t see how the discussion about the MPXV reservoir follows naturally after the section Mpox Cases in the Post-COVID-19 Pandemic Period. Also, I don’t see how Animal Models Employed in the Study of Mpox Infection follows naturally after Clinical Features. 

We thank the reviewer for this suggestion. However, we cannot change the order of section within the manuscript because the section order was structured carefully to ensure that the content flows logically and can effectively communicate to the reader. Changing the order of sections would significantly impact the rationale and objective of manuscripts.

Here´s an explanation of why altering the section order would be detrimental:

  1. Introduction: It provides background information on the topic and establishing its significance. It sets the stage for the comprehensive overview and introduces the reader to the disease.
  2. Public Health Importance: This section explores the broader implications and significance of human monkeypox from a public health perspective. It discusses the impact on healthcare systems, and global health. Having this section following the introduction builds upon foundational knowledge and emphasizes the importance of addressing the disease comprehensively.
  3. Mpox cases in the Post-COVID-19 Pandemic Period: This section specifically focuses on cases and trends of mpox post the COVID-19 pandemic period. By placing it after the “public health importance section”, the reader gains a contextual understanding of how the pandemic may have influenced the prevalence, transmission and response to human monkeypox.
  4. Animal Reservoirs: This section delves into the animal species that serve as reservoirs for MPVX. Having this section next provides an overview of the disease and its impact where the reader is now prepared to understand the role of animal reservoirs in epidemiology and transmission dynamics.
  5. Viral attachment and pathogenesis: This section explores the mechanism by which MPVX attaches to host cells and progresses. Placing it after the “animal reservoirs” section allows for a logical progression of information, enabling the connection of the disease´s biology with its broader context.
  6. Immunopathophysiology: This section focuses on the immune responses and pathological processes triggered by human monkeypox infection. Having this section followed by the “viral attachment and pathogenesis section” provides a deeper understanding of the disease´s impact on the host´s immune system and overall health.
  7. Modes of Transmission: In this section, we outline the various ways in which human monkeypox spreads among individuals. Placing it after the previous sections “Viral attachment and pathogenesis” and “Immunopathophysiology” helps comprehend the factors that contribute to transmission and the implications for disease control and prevention.
  8. Laboratory diagnosis: We discuss diagnostic methods used to identify MPVX. Having this section after “Viral attachment and pathogenesis”, “Immunopathophysiology” and “Modes of Transmission” allows the reader to understand the clinical manifestations and the rationale behind different diagnostic approaches.
  9. Clinical features: This section explore the signs, symptoms, and clinical presentation of mpox. Placing it after “laboratory diagnosis” ensures that readers are familiar with the diagnostic methods, before delving into the diseases’ clinical aspects.
  10. Animal Models Employed in the Study of Mpox Infection: This section focuses on the animal models used to study human monkeypox infection. By following the previous sections “clinical features” and “laboratory diagnosis”, it provides a logical progression that allows readers to understand how animal models contribute to the broader understanding of the disease.
  11. Cross-Immunity and Vaccines: This section discusses the concept of cross-immunity and development of vaccines for human monkeypox. Placing it after the animal models sections connects the research of animal models with the potential preventive strategies, and highlights the importance of immunization.
  12. Treatment and Prevention: This section explores prevention strategies and the available treatment options for human monkeypox. Placing it near the end ensures that readers have a comprehensive understanding of the disease´s epidemiology, pathogenesis, diagnosis, and potential interventions, before addressing treatment and prevention.
  13. Concluding remarks: This section provides a summary of the key findings, emphasizing the significance of the comprehensive overview, and suggests future directions for research and intervention. These reflect on the entirety of the manuscript and its implications.

Based on our criteria, the order of manuscripts sections in "Human Monkeypox: A comprehensive overview of epidemiology, pathogenesis, diagnosis, treatment, and prevention strategies" was designed to provide a cohesive narrative that allow readers to grasp the disease’s various aspects in a structured manner. Altering the section order would disrupt the flow of information, impede the understanding, and undermine the manuscript’s sense and objective that was designed and developed by the authors. Additionally, reviewer 1 and 2 have not expressed any disagreement or comments about the section order. On the contrary, they have expressed agreement with our revised version with their comments incorporated.

The text still needs perusal for English.  I list a few problems. 

We appreciate this comment. MDPI have checked the manuscript again and we received an English editing certificate in June 2023, english-65307. The English is now clear, consistent, and grammatically correct in the new version of the manuscript. Any concerns could be further answered via email (authorservices@mdpi.com). Additionally, a colleague in this research field who is a native English speaker over our paper.

1) Introduction, line 36: “Human monkeypox is a zoonotic infection…”  The authors refer to mpox.  However, the term “mpox” is first mentioned only in line 58, without explanation. In line 59, the origins of the ten “mpox” are clarified. This clearly needs to be corrected.

We appreciate this general observation. However, we cannot accommodate this suggestion for the following reason: The terms we used in the manuscript were according to the nomenclature used by the World Health Organization. The term “mpox” is used to refer to the monkeypox disease, while the term MPXV is an acronym that stands for Monkeypox Virus, which specifically refers to the virus responsible for causing monkeypox disease or mpox.

Caption of Fig 1, line 108: “Epidemiological curves of mpox in the world. (A and B) Global cases and deaths reported by epidemiological weeks, respectively…” What are epidemiological curves? What are epidemiological weeks?

We appreciate this comment. We edited this figure legend and it is now: “Confirmed mpox cases and deaths in the world (A and B). Global cases and deaths, respectively (data represent the cases in the week) (C). Top ten countries with confirmed cases and (D) top ten countries with reported deaths. Data are derived from the WHO (accessed on 29 May 2023). (page 3, lines 202-205).

3) line 451: “Although mpox is usually resolved spontaneously, recovering without treatment in a few weeks…” The patient recovers, not the disease. I guess the authors wish to convey that mpox often clears without treatment. 

We appreciate this general observation. We edited this sentence and is now: “Although mpox often clears without treatment, the severity and risk of mortality can be influenced by factors such as the clade of the infecting mpox strain, the route of exposure, and the immune status of the person” (page 10, lines 724-726).

Round 3

Reviewer 3 Report

I still do not understand the logic and the flow of the current sectioning of the paper. Listing a few comments about each section does not help the argument. The fact that you got confirmation from reviewers 1 and 2, does not help the argument either. My comments in my previous report still stand. 

I do not understand the reason for your confession about how you improved the English of the manuscript. I have the text in front of me and I can clearly see that it is still not free of typos. Obviously, the responsibility for the quality of the English language in the manuscript rests with the authors. Here are again a few problems:

1) My previous comment: 

Introduction, line 36: “Human monkeypox is a zoonotic infection…”  The authors refer to mpox.  However, the term “mpox” is first mentioned only in line 58, without explanation. In line 59, the origins of the ten “mpox” are clarified. I mentioned that this needs to be corrected. 

Your answer:

We appreciate this general observation. However, we cannot accommodate this suggestion for the following reason: The terms we used in the manuscript were according to the nomenclature used by the World Health Organization. The term “mpox” is used to refer to the monkeypox disease, while the term MPXV is an acronym that stands for Monkeypox Virus, which specifically refers to the virus responsible for causing monkeypox disease or mpox.

My reply is that “Human monkeypox” no longer belongs to the current WHO nomenclature and should be avoided to start with. I see no reason whatsoever that the paper does not stick thoroughly and entirely with the WHO nomenclature. I believe that the nomenclature requires no explanation, but that rests with the authors. Consider the before to last sentence in the Abstract: “The material presented here provides a comprehensive understanding of MPXV as a disease while…”. MPXV is not a disease. 

2) First sentence of the Abstract:

“Monkeypox virus (MPXV) is an emerging zoonotic virus that belongs to the Orthopox-virus genus and presents clinical symptoms similar to those of smallpox…” It is obvious that the virus may determine symptoms but it does not present any symptoms. Next sentence in the Abstract “generate” should be “generates”.

3) Caption of Figure 1: “data represent the cases in the week” should be “data represent cases per week”.

The manuscript requires perusal for the English language.